# Two New Fossil Sawflies of Pamphiliidae (Hymenoptera: Symphyta) from the Mesozoic of Northeastern China ^†^

**DOI:** 10.3390/insects13050402

**Published:** 2022-04-22

**Authors:** Jialiang Zhuang, Chungkun Shih, Mei Wang, Dong Ren

**Affiliations:** 1College of Life Sciences and Academy for Multidisciplinary Studies, Capital Normal University, Beijing 100048, China; zhuangjialiang2022@163.com (J.Z.); chungkun.shih@gmail.com (C.S.); 2Key Laboratory of Forest Protection of National Forestry and Grassland Administration, Institute of Forest Ecology, Environment and Nature Conservation, Chinese Academy of Forestry, Beijing 100091, China; 3Department of Paleobiology, National Museum of Natural History, Smithsonian Institution, Washington, DC 20013-7012, USA

**Keywords:** Juralydinae, Cephalciinae, Pamphiliinae, insect fossil, taxonomy, tarsal claw

## Abstract

**Simple Summary:**

Two new species and one new specimen of *Scabolyda* (Pamphiliidae) are described from the Mesozoic of northeastern China. Structures of antennae, genitalia, and legs, especially the hind tarsal claw, are preserved. The documentation of these new structures helps to link extant and fossil taxa of Pamphiliidae.

**Abstract:**

Two new species of Pamphiliidae, *Scabolyda latusa* sp. nov. and *Scabolyda tenuis* sp. nov. are described and illustrated from the late Middle Jurassic Jiulongshan Formation and the Lower Cretaceous Yixian Formation of northeastern China, respectively. A new specimen of *Scabolyda orientalis* Wang, Rasnitsyn, Shih and Ren, 2014 with distinct male genitalia is documented for the first time. Based on the specimens with new and distinct structures of legs, antennae, and genitalia, the morphological characters of *Scabolyda* are supplemented: antenna with ca. 13–14 flagellomeres; fore leg with tibia without pre-apical spur; hind leg nearly 0.6 times as long as the body, hind tarsal claw without setae and its inner tooth not developed. In addition, the tarsal claw characteristics found in the new species may suggest *Scabolyda* has a closer relationship with Cephalciinae, rather than with Pamphiliinae.

## 1. Introduction

Pamphilioidea, a small superfamily with phylogenetic proximity to Xyeloidea and Tenthredinoidea, was suggested as the sister group to all remaining Hymenoptera in a recent study [1]. Originally, two extinct families (Praesiricidae and Xyelydidae) and two extant families (Pamphiliidae and Megalodontesidae) were attributed to Pamphilioidea [2,3,4]. Wang et al. [5] conducted a phylogenetic analysis of Pamphilioidea based on morphological characters and DNA sequence data. It was indicated that Praesiricidae is not monophyletic, and they proposed that the paraphyletic Praesiricidae be synonymized under Megalodontesidae. Later, Wang et al. [6] described two specimens of *Mirolyda hirta* Wang, Rasnitsyn and Ren, 2017, in Mirolydidae, with a unique combination of features: forewing with Sc developed, 1-Rs nearly as long as 1-M, M+Cu straight, antenna with the first flagellomere homonomous, and body surface with long and thin setae. Based on the results of the phylogenetic analysis, they suggested Mirolydidae be considered as a new family in Pamphilioidea. Pamphilioidea now comprises two extinct families, Xyelydidae and Mirolydidae, and two extant families, Pamphiliidae and Megalodontesidae.

The family Pamphiliidae consists of three subfamilies, Pamphiliinae Cameron 1890, Cephalciinae Benson 1945, and Juralydinae Rasnitsyn 1977 [7]. The subfamilies of Pamphiliinae and Cephalciinae have more than 300 extant species [8]. Although they are close relatives, it is difficult to distinguish them with the naked eye. They have adapted to different types of vegetation: the larvae of Cephalciinae mainly feed on coniferous plants but those of Pamphiliinae feed on angiosperms [9,10]. Additionally, Cephalciinae also contain some fossil taxa, including species of *Tapholyda caplani* Cockerell, 1933 (the Oligocene of USA and the Miocene of Russia), *Acantholyda erythrocephala* Linnaeus, 1758 (the Miocene of France) and possibly *Acantholyda ribesalbesensis* Peñalver, and Arillo, 2002 (the Miocene of Spain) [4,11,12,13,14].

So far, a total of five species in three genera are assigned to the extinct subfamily Juralydinae: *Atocus defessus* Scudder, 1892 and *Atocus cockerelli* Rohwer, 1908 (the terminal Eocene of USA), *Juralyda udensis* Rasnitsyn, 1977 (Uda Formation, the Upper Jurassic of Russia), *Scabolyda orientalis* Wang, Rasnitsyn, Shih and Ren, 2014 (Jiulongshan Formation, the late Middle Jurassic of China), and *Scabolyda incompleta* Wang, Rasnitsyn, Shih and Ren, 2014 (Yixian Formation, the Lower Cretaceous of China) [4,7,15,16].

In addition, *Ulteramus republicensis* Archibald and Rasnitsyn 2015 (the early Eocene of USA), a specimen only preserved with the forewing, has not been placed in any subfamily due to the distinct character of Sc2 joining R distal to 1-Rs and limited information in the specimen [17].

Recently, we collected four new fossil specimens of Pamphiliidae from the Jiulongshan Formation and Yixian Formation. After detailed examinations, we consider these specimens belong to a previously erected extinct genus *Scabolyda* Wang, Rasnitsyn, Shih and Ren, 2014 [16]. Based on these well-preserved specimens, the structures of legs and antennae of extinct and extant species of Pamphiliidae are examined and compared. These new structures can enhance our understanding of the Mesozoic taxa in Pamphiliidae and provide more comprehensive evidence to compare and study the early evolution of Pamphiliidae.

## 2. Materials and Methods

The fossil specimens were collected from the Jiulongshan Formation, Daohugou Village, Wuhua Township, Ningcheng County, Chifeng City, Inner Mongolia, China and the Yixian Formation, Huangbanjigou, Beipiao City, Liaoning Province, China. All of them (one new specimen, two holotypes and one paratype) described in the paper are housed at the Key Laboratory of Insect Evolution and Environmental Changes, College of Life Sciences and Academy for Multidisciplinary Studies, Capital Normal University, Beijing, China (CNUB; Dong Ren, Curator).

The specimens were examined and photographed under Nikon SMZ 25 dissecting microscope with an attached Nikon DS-Ri2 digital camera system, either dry or wetted with 95% ethanol. The line drawings were prepared using Adobe Illustrator CC 2022 and Adobe Photoshop CC 2021. The wing venation nomenclature used in this study is modified from Rasnitsyn [2,3].

## 3. Results

Systematic paleontology.

Order Hymenoptera Linnaeus, 1758.

Suborder Symphyta Gerstaecker, 1867.

Superfamily Pamphilioidea Cameron, 1890.

Family Pamphiliidae Cameron, 1890.

Subfamily Juralydinae Rasnitsyn, 1977.

Genus *Scabolyda* Wang, Rasnitsyn, Shih and Ren, 2014.

Type species. *Scabolyda orientalis* Wang, Rasnitsyn, Shih and Ren, 2014.

Species included. *Scabolyda orientalis* Wang, Rasnitsyn, Shih and Ren, 2014; *Scabolyda incompleta* Wang, Rasnitsyn, Shih and Ren, 2014; *Scabolyda latusa* sp. nov. and *Scabolyda tenuis* sp. nov.



*Scabolyda orientalis* Wang, Rasnitsyn, Shih and Ren, 2014.

urn:lsid:zoobank.org:act:21407ED8-78B8-44F8-AFEE-B46DC1B887D0

New specimen. CNU-HYM-NN2022101p/c (part and counterpart).

Locality and horizon. Jiulongshan Formation; Daohugou Village, Shantou Township, Ningcheng County, Inner Mongolia, China (41°18.979′ N, 119°14.318′ E); latest Middle Jurassic of late Callovian age.

Description (Figure 1). Male. Head flat and oval, covered with small punctations. Mandibles developed, eyes medium-sized, and three ocelli triangularly arranged at the center of head. Postocellar furrow located behind the ocellus, lateral furrows sub-parallel, and extending forward between the antennal sockets. Antenna with ca. 15–16 segments, the basal part vaguely visible, the visible parts of flagellomeres gradually shortened and their length-width ratio (excluding the first flagellomere) from 3 times to 1.9 times toward the apex, except for the last flagellomere slender and 3.3 times as long as wide.

Thorax width equal to head width, pronotum trapezoid. Mesothorax with mesoprescutum nearly as large as mesoscutellum. Cenchrus clear and distinct, metascutellum oblong. All three pairs of legs covered with short setae. Fore leg with femur slightly wider than tibia, tibia lacking the pre-apical spur and 1.2× as long as femur (Figure 1C). Mid leg with tibia partly preserved, one pre-apical spur visible distad middle part (Figure 1D). Hind leg with coxa trapeziform, trochanter, and trochantellus clearly visible. Coxa nearly 1.3× as long as wide. Femur inflated and the middle part less than twice as wide as the basal part; tibia long and slender, nearly 1.4× as long as femur (Figure 1E). Tarsus with the first segment 2.3× as long as the second segment, apical tarsomere nearly as long as the second segment, arcus small, nearly round at the apex of pretarsus, its diameter equal to the basal width of claw in length and 0.6× the width of apical tarsomere. Claw simplified and lacking inner tooth, its basal part slightly widened (Figure 1F).

Abdomen with seven segments; male genitalia distinct, gonocoxa inverted trapeziform, gonostylus triangular, full of short setae, and nearly as long as the gonocoxa in ventral view, digitus valviform, and covered with setae, and penial valve blunt at the apex (Figure 1G,H).

Forewing with pterostigma sclerotized. Sc bifid, Sc2 joining R proximal to 1-Rs. 1-Rs nearly half of 1-M in length, M+Cu bent. Angle of 1-M and 1-Cu nearly 116°. Length proportions of vein 1-M: Rs+M: 1-Cu: 2-M = 0.51: 0.75: 0.49: 0.45. 2r-rs twice as long as 1r-rs, 2r-m almost parallel to 2r-rs. 3-Rs sharply curved, 4-Rs short, cu-a located proximal to the middle of cell 1 mcu, 1 m-cu 0.3× as long as 3-Cu. Cell 1 mcu ca. 1.3× as long as wide, cell 2a 2.7× as long as wide. Hind wing with cell r and rm vaguely visible.

Measurements (in mm). Body length (excluding antenna) 12.27; head width 2.95 and length 1.79; antenna 5.37 in length; forewing length 7.94, width 3.33; fore femur length 1.55; fore tibia length 1.81; hind leg length 7.83 (coxa length 0.48; femur length 1.99; tibia length 2.86; basal tarsomere length 0.79).

Remarks. The new specimen has the common characters of Pamphiliidae: forewing with Sc developed, R sinuate and not straight before RS base, M+Cu angularly bent. Furthermore, the specimen can be assigned to the *Scabolyda orientalis* in Juralydinae for having Sc bifid, 1-Rs nearly half length of 1-M, Rs+M not reaching twice length of 2-M, M+Cu without extra stub, cell 1 mcu 1.3× as long as wide in forewing.

Wang et al. [16] made a detailed description of *S. orientalis* based on four specimens. However, the structures about legs and genitalia were not described due to the lack of preservation of corresponding parts in these four specimens. For this new specimen CNU-HYM-LB2022101p/c, the description of both structures is added. Comparing *S. orientalis* with the extinct groups, there are many similarities for these structures, except for the lack of inner tooth in the claw of hind leg for *S. orientalis*, which is different from all other extant pamphiliids. Additionally, there is no pre-apical spur on the fore tibia for *S. orientalis*, which is similar to that of extant taxa, with the exception of genus *Acantholyda* with a pre-apical spur on the fore tibia [9].



*Scabolyda latusa* Zhuang, Shih, Wang and Ren sp. nov.

urn:lsid:zoobank.org:act:E1FE78EA-6F56-4547-93C9-09E2D906DF37

Material. Holotype, CNU-HYM-NN2022102p/c (part and counterpart; Figure 2); Paratype, CNU-HYM-NN2022103p/c (part and counterpart; Figure 3).

Locality and horizon. Jiulongshan Formation; Daohugou Village, Shantou Township, Ningcheng County, Inner Mongolia, China (41°18.979′ N, 119°14.318′ E); latest Middle Jurassic of late Callovian age.

Etymology. The specific name “*latusa*” (latus = wide) is a Latin word, referring to the broad antenna of this species.

Diagnosis. Large body size (nearly 20 mm in length). Body surface with obvious punctations, especially on head and thorax. Antenna with the first flagellomere nearly 4× as long as the second flagellomere; the ratio of length to width of the following flagellomere nearly two; forewing with vein 4-Rs present and short. Hind tarsal claw with inner tooth small, located submedial.

Description (Figure 2 and Figure 3). Holotype. Female. Head very pale with black areas on part of postocellar area, frons, and gena; many obvious punctations surrounding the ocellus and around the postocellar area and gena. The whole antenna pale brown, except for the basal scape blackish. Thorax color pale with the lateral mesothorax and partial metathorax black; mesonotum and metanotum densely covered with punctations. Hind leg pale brown except for the coxa blackish. The whole abdomen and forewing pale brown (Figure 2).

Head massive and flat, its width about 1.2× length; postocellar furrow and lateral furrow visible; middle fovea and lateral fovea indistinct; frons weakly raised. Eyes medium-sized and nearly half the length of head; temple obviously raised. Antenna ca. 2.3× as long as the width of head, and the left antenna 15-segmented; scape 3.9× as long as pedicel, and 0.9× as wide as apical pedicel, its length shorter than the first flagellomere; pedicel 0.2× as long, and almost as wide as the first flagellomere; the first flagellomere 4.9× as long as wide and ca. 3.9× the length of the second flagellomere; the second flagellomere 1.6× as long as wide and slightly shorter than the third flagellomere, each of the remaining flagellomeres not more than 2.3× as long as wide and gradually thinning toward the apex.

Thorax with pronotum 0.8× as wide as head; mesothorax wider than head, with notaulices, parapsidal suture, mesoprescutum, and mesoscutellum visible. Mesoprescutum about half of mesoscutum in width, and larger than mesoscutellum. Metathorax with metapostnotum and parapsis visible, cenchrus indistinct. Leg with hind leg visible, full of setae, and its length longer than abdomen; coxa large, inverted trapeziform, not reaching half of abdomen; trochanter and trochantellus visible. Femur markedly thick and with five spurs obviously present; middle part of femur thicker than basal part but not reaching twice. Tibia thin and nearly 1.3× as long as femur. Tarsus partly visible; claw with two teeth and the inner tooth small (Figure 2E).

Abdomen with seven segments preserved. Ovipositor as preserved typical of Pamphilioidea: very short, with stylets (valvula 1 and valvula 2) widely separated basally and meeting only at the apex (Figure 2D).

Forewing with pterostigma slender and sclerotized, and about 3.6× as long as wide. Sc bifid; Sc2 shorter than Sc1 and entering R before 1-Rs; R bent strongly near the middle part. 1-Rs approximately 0.5× as long as 1-M and nearly as long as 1r-rs, inclined toward wing apex; 1r-rs ca. 0.4× as long as 2r-rs. 2r-m parallel to 2r-rs, meeting Rs slightly distal to 2r-rs. 3r-m inclined toward wing apex and meeting 5-Rs nearly at 121°; Rs+M 1.3× as long as 1-M; 1-M meeting 1-Cu at an angle of 127°; M+Cu obviously bending and without stub around the corner. Cell 1 mcu ca. 1.3× as long as wide, with cu-a located distad the middle of cell. 2-M and Rs+M nearly equal in length; 3-Cu at least 2.8× as long as 1 m-cu; 2 m-cu curved near its middle. Cell 2a ca. 3.5× as long as wide. Hind wing with Sc present, reclined and longish. 1-Rs nearly as long as 1-M. Both 1A and 2A bent, and 2A more curved than 1A.

Paratype. Female. Head flat, full of punctations, and significantly raised on both sides. Mandibles well-developed and mostly occupying more than half width of the head, with a large submedial inner tooth on both sides (Figure 3C,D). Eyes medium-sized, nearly half length of head. Lateral furrow vaguely visible.

Thorax with punctations. Mesothorax with mesoprescutum, mesoscutellum, and mesoscutellar appendage visible. Cenchrus slightly smaller than mesoscutellar appendage. Hind leg with coxa, trochanter and femur preserved, coxa inverted trapezoid and nearly reaching the posterior margin of the second tergite. Abdomen with eight segments, ovipositor short and incompletely preserved.

Forewing nearly reaching the seventh tergite. Pterostigma long and sclerotized. Sc developed and forked; Sc2 joining R proximal to 1-Rs and the distance between them about its own length. Radial vein obviously curved at the middle part. 1-Rs short and 0.6× as long as 1-M. Angle of 1-M and 1-Cu nearly 129°. Length proportions of vein 1-M: Rs+M: 1-Cu: 2-M = 1.06: 1.28: 1.32: 1; cu-a located distad the middle of cell 1 mcu, 1 m-cu 0.3× as long as 3-Cu. Cell 1 mcu 1.3× as long as wide.

Measurements (in mm). Holotype. Body length (excluding antenna) 19.64; head width 4.04, length 3.41; antenna 7.65 in length; the first flagellomere 1.53 in length; forewing length 13.52, width 5.83; hind leg length 12.66 (coxa length 0.79; femur length 3.23; tibia length 4.22); abdomen 9.64 in length. Paratype. Body length 20.37; head width 4.27, length 3.01; forewing at least 14.08 in length.

*Scabolyda tenuis* Zhuang, Shih, Wang and Ren sp. nov.

urn:lsid:zoobank.org:act:9823F1EC-A545-4450-A2ED-36ECCD88D511

Material. Holotype, CNU-HYM-LB2022104.

Locality and horizon. Yixian Formation; Huangbanjigou, Chaomidian Village, Beipiao City, Liaoning Province, China; Lower Cretaceous.

Etymology. The Latin name “*tenuis*” meaning thin, referring to the long and slender flagellomere of this species.

Diagnosis. Body surface without punctations. Flagellomeres, except for the first flagellomere, slender and long, more than three times as long as wide. Forewing with Sc1 and Sc2 nearly equal in length; 2r-rs and 2r-m almost in a line, 4-Rs absent.

Description (Figure 4). Holotype. Sex unknown. Head, thorax and abdomen brown and without punctations. The basal scape color dark brown; the rest part of scape and the first flagellomere brown; the other flagellomeres light yellow. Veins of forewing and hind wing with color ocher and without any wrinkles (Figure 4).

Head massive and flat, nearly 1.6× as wide as long; postocellar furrow and lateral furrow visible. Eye medium-sized and its bottom part below the postocellar furrow. Antenna partly preserved, at least ten segments; scape 0.7× as long as the first flagellomere, nearly as wide as pedicel; pedicel trapezoidal. The first flagellomere lightly dilated, 4.5× as long as wide, and nearly as long as the sum of next three segments; the second flagellomere 3.6× as long as wide; other flagellomeres left (some preserved incompletely) ca. 3.5× as long as wide.

Thorax slightly wider than head; pronotum partly preserved; mesoscutum faintly visible, nearly inverted triangle. Mesoprescutum larger than mesoscutellum. The remaining part of the thorax not discernible. Abdomen with eight segments preserved; the first segment divided medially. Genitalia poorly preserved, thus sex unknown.

Forewing with pterostigma long and sclerotized, ca. 3.7× as long as wide. Sc forked and its main vein located nearly at the middle of costal area; Sc2 almost as long as Sc1, and both of them entering vein R or C before the base of 1-Rs (Figure 4D); 1-Rs inclined apically and about half the length of 1-M; 1r-rs slightly longer than 1-Rs and 0.5× as long as 2r-rs; 2r-m almost in alignment with 2r-rs; 3r-m inclined toward wing apex. M+Cu bent and the angle at the bending about 152°. Length proportions of vein 1-M: Rs+M: 1-Cu: 2-M = 0.59: 0.85: 0.86: 0.53. Angle of 1-M and 1-Cu nearly 106°. Cell 1 mcu length about 1.5× width; cu-a situated at the middle of cell 1mcu. Cell 2a nearly 2.9× as long as wide. Hind wing preserved incompletely; 1-Rs 0.7× as long as 1-M; cu-a slightly bent and 1.2× as long as 1-M; 1A and 2A curved.

Measurements (in mm). Holotype. Body length (excluding antenna) 10.32; head width 3.16 and length 1.96; antenna at least 5.28 in length; third antennal segment 1.31 in length; forewing length 8.4, width 3.33.

Remarks. The two new species are assigned to Pamphiliidae for the same reasons in the remarks for the new specimen of *Scabolyda orientalis* above. Both new species can be further attributed to the *Scabolyda* based on the combination of these morphological characters: antenna with the first flagellomere nearly three times as long as the second flagellomere; forewing with pterostigma slender, Sc developed and forked, Sc2 entering R proximal to 1-Rs, 1-Rs nearly half of 1-M in length, angle between 1-M and 1-Cu over 90°, 2r-rs close to the apical pterostigma, and 3r-m inclined toward the wing apex.

With the descriptions of two new species, the Mesozoic pamphiliids are diversified and the relevant character structures are expanded, such as antennae, legs, and genitalia. In order to better distinguish the four species of *Scabolyda*, we selected the key characters and established the relevant key to the identification of these four species of *Scabolyda*.




**Key to the species of *Scabolyda***
1. Body surface with obvious punctations, especially on head and thorax; most length-width ratio of flagellomere ca. 2–2.5 times..................................................................................................................... 2 Body surface without punctations; most length-width ratio of flagellomere ca. 3–4 times................ 3 2. Forewing with 4-Rs present; antenna with the first flagellomere nearly four times as long as the second flagellomere................................................................................................ *Scabolyda latusa* sp. nov. Forewing with 4-Rs present; antenna with the first flagellomere nearly three times as long as the second flagellomere....................................... *Scabolyda orientalis* Wang, Rasnitsyn, Shih and Ren, 2014 3. Forewing with Sc close to vein C, 4-Rs present, Rs+M nearly twice length of 2-M; antenna with the second flagellomere shorter than the third flagellomere..................................................................... ........................................................................ *Scabolyda incompleta* Wang, Rasnitsyn, Shih and Ren, 2014 Forewing with Sc located at the middle part of cell C, 4-Rs absent, Rs+M shorter than twice length of 2-M; antenna with the second flagellomere nearly as long as the third flagellomere........................ .................................................................................................................................. *Scabolyda tenuis* sp. nov. 


## 4. Discussion

As one of the basal groups in Symphyta, Pamphiliidae have very few Mesozoic fossil record and until now, there are only three species [7,14]. Some minute but important characters used for the classification of extant taxa are difficult to be preserved in compression fossils, such as the arrangement of the inner tooth in the tarsal claw, tibial spines, and spurs, either membranous or completely sclerotized at the tips, etc. [9,10]. In this study, we documented two new fossil species with well-preserved antennae and legs, which provided clearly visible characters to address the issues of unknown minute characters. With the addition of these features, it will be helpful to classify and identify species of Pamphiliidae with more than just venational characters commonly used for fossils.

The newly documented characters of antennae and legs in *Scabolyda* are as follows: antenna with ca. 13–14 flagellomeres, fore leg with tibia lacking pre-apical spur, mid leg with tibia having at least three spurs, hind leg nearly 0.6 times as long as the body, hind tibia with five spurs, the inner tooth of hind tarsal claw not developed and the claw without setae. Herein, we suggest treating these antenna and leg characters as diagnostic characters for *Scabolyda* as well.

After comparisons, the antenna is different at the species level and the leg, especially the tarsal claw of hind leg, is different at the subfamily level. Antennal morphology is an important character to distinguish different taxa in Pamphilioidea. Compared with other families, the scape and the first flagellomere are normal, and the length ratio between the first flagellomere and the second flagellomere is relatively stable (nearly 2–3 times) in Pamphiliidae [18]. However, the antennae for species of *Scabolyda* have diversified forms: the flagellomere is relatively widest in *S. latusa* sp. nov.; the first flagellomere is shortest in *S. incompleta*; length–width ratio of flagellomere (except the first flagellomere) is ca. 2–2.5 times in the Middle Jurassic Daohugou groups (Figure 5A,B) and 3–4 times in the Lower Cretaceous Yixian groups (Figure 5C,D). The main length–width ratio of flagellomeres and comparison of lengths are shown in Table 1. In spite of the differences, the possibility cannot be excluded that some reported species might belong to the same species for the existence of sexual dimorphism which is common in Hymenoptera [19]. In addition, some documented specimens with genital organs poorly-preserved made the comparison and judgment even more difficult.

The claw, usually located at the apex of the leg, plays an important role in support and sensing [20,21,22]. Ermolenko [23] used to study the tarsal claw structure in representative Symphyta and the shape of claw was later widely used as a taxonomic character. The two extant subfamilies of Pamphiliidae have different types of tarsal claw. In Cephalciinae, the claw is generally long and slender, which has a minute and perpendicular inner tooth, located submedial (Figure 6A). While in Pamphiliinae, the claw is bifurcate and the basal part obviously broadened. The inner tooth is pointing to the apex of the claw and longer than its basal width (Figure 6B) [17]. For the extinct *Scabolyda* in Juralydinae, the type of claw is very similar to that of Cephalciinae. The whole claw is slightly bent on the inner side and has a small and perpendicular tooth in the species *S. latusa* sp. nov. Moreover, there is no such minute tooth in *S. orientalis* (Figure 6C), which may be the more basal (ancestor) condition of the claw and similar to that of the genus *Xyela* (Family Xyelidae) [21]. Moreover, the lengths of the setae covering the claws in extant groups often reach at least the lengths of the setae on their apical tarsomeres, while the setae were not developed or possibly absent in the extinct *Scabolyda*, as shown in specimens. The lack of setae on the claw of *Scabolyda* suggests that these extinct sawflies might have held less tarsal sensing efficiency when compared with the extant sawflies.

The characters of the claw found in these newly reported specimens suggest that *Scabolyda* might have a closer relationship with Cephalciinae, rather than with Pamphiliinae. At present, the species of extant Cephalciinae live in habitats full of conifers [9]. Furthermore, we explore the possibility that the Pamphiliidae also lived on coniferous plants in the Mesozoic. According to previous research, gymnosperms were dominant and most types of conifers were present in the Jurassic, peaking in the Late Jurassic to the Early Cretaceous [24], which happened to be the age *Scabolyda* reported. Additionally, a variety of conifer fossils have also been reported in the Daohugou Biota and Yixian Biota [25,26,27]. Therefore, based on the potential habitats in the Jurassic and the similar structures between Cephalciinae and *Scabolyda*, we suggest that the Jurassic species of *Scabolyda* might have lived on conifers as well. Some of them might have further adapted to conifers and established strong bonds in later evolution.

## 5. Conclusions

Due to the limitation on the preservation of fossil material, the classification of the extinct Pamphiliidae has previously mainly been based on wing venation, such as vein Sc developed or not, the relative length of 1-Rs, the width of the pterostigma, etc. However, the extant taxa are often classified through more diverse aspects, such as the surface colors, the spurs, and spines in tibia, the number of teeth in the mandible, and so on. This information asymmetry makes it hard to compare fossil and extant taxa in some characteristics, leading to weakened inter-relationships between them.

Although the fossil record of Pamphiliidae is currently small, researchers have tried to add features other than wing venation to the diagnostic characters. Rasnitsyn [4] added the shape of head, the segmentation of antenna, the width of the femur and the hollow of the last sternite in the diagnoses of genera of *Atocus* and *Tapholyda*. Wang et al. [16] described the characteristics of *Scabolyda* regarding the first to third flagellomeres and the scutum in mesothorax. Obviously, these characters are far from sufficient. In this study, we added the characteristics of antennae and legs in *Scabolyda*: antenna with ca. 13–14 flagellomeres; fore leg with tibia without pre-apical spur; hind leg nearly 0.6 times as long as the body, hind tarsal claw without setae, and its inner tooth small or absent.

We observed some differences in antennae among different species of *Scabolyda*, which may be due to the sexual dimorphism. The structures of the tarsal claw are rarely preserved and, thus, have not been reported in Symphyta fossils before. The hind tarsal claw of *Scabolyda* lacks setae and has a small and perpendicular inner tooth or is absent, which is morphologically more similar to that of Cephalciinae.

## Figures and Tables

**Figure 1 insects-13-00402-f001:**
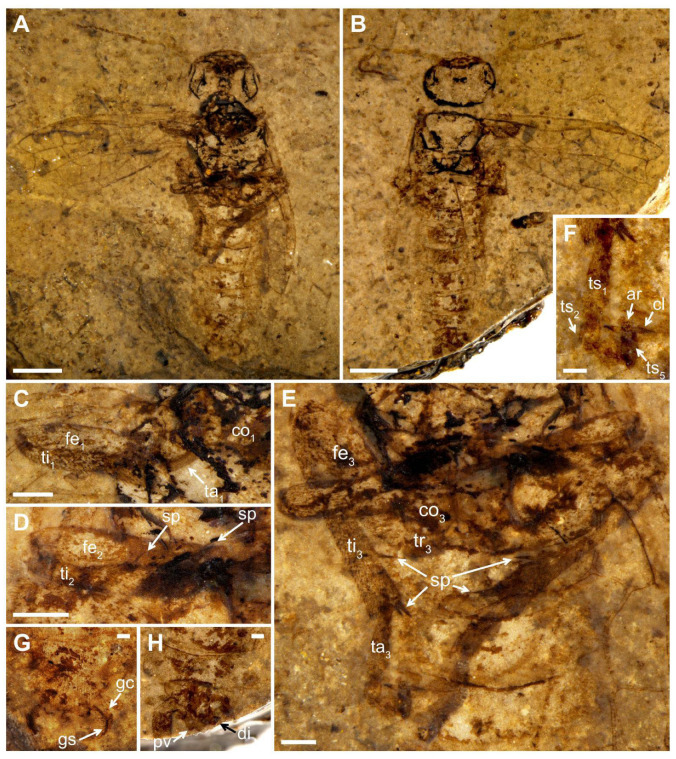
*Scabolyda orientalis* Wang, Rasnitsyn, Shih and Ren, 2014, New specimen (CNU-HYM-NN2022101p/c), male. All photos were taken wetted with 95% ethanol. (**A**) Ventral view of body as preserved. (**B**) Dorsal view of body as preserved. (**C**) Fore leg. (**D**) Mid leg. (**E**) Hind leg. (**F**) Hind tarsus. (**G**) Ventral view of genitalia as preserved. (**H**) Dorsal view of genitalia as preserved. Symbols: co_1_, co_3_ = fore and hind coxa; tr_3_ = hind trochanter; fe_1_, fe_2_, fe_3_ = fore, mid and hind femur; ti_1_, ti_2_, ti_3_ = fore, mid, and hind tibia; sp = spur; ta_1_, ta_3_ = fore and hind tarsus; ts_1_–ts_5_ = 1st–5th tarsal segments; cl = claw; ar = arcus; gc = gonocoxa; gs = gonostylus; pv = penial valve; di = digitus. Scale bars: 2 mm in (**A**,**B**); 0.25 mm in (**C**–**E**); 0.25 mm in (**F**–**H)**.

**Figure 2 insects-13-00402-f002:**
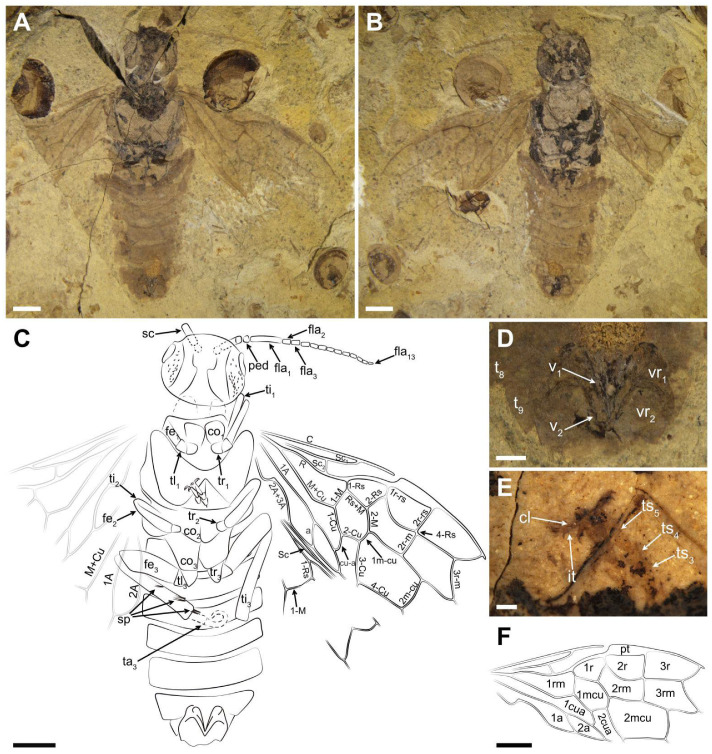
*Scabolyda latusa* sp. nov., Holotype (CNU-HYM-NN2022102p/c) female. Photo 2E was taken wetted with 95% ethanol. (**A**) Ventral view of body as preserved. (**B**) Dorsal view of body as preserved. (**C**) Line drawing of ventral view. (**D**) Ovipositor. (**E**) Mid tarsus. (**F**) Line drawing of forewing. Symbols: sc = scape; ped = pedicel; fla_1_, fla_2_, fla_3_, fla_13_ = 1st, 2nd, 3rd, and 13th flagellomere; co_2_ = mid coxa; tr_1_, tr_2_ = fore and mid trochanter; tl_1_, tl_3_ = fore and hind trochantellus; t_8_, t_9_ = 8th and 9th tergum; v_1_, v_2_ = 1st and 2nd valvifer; vr_1_, vr_2_ = 1st and 2nd valvifer; it = inner tooth; pt = pterostigma. Scale bars: 2 mm in (**A**–**C**,**F**); 0.5 mm in D; 0.25 mm in (**E**).

**Figure 3 insects-13-00402-f003:**
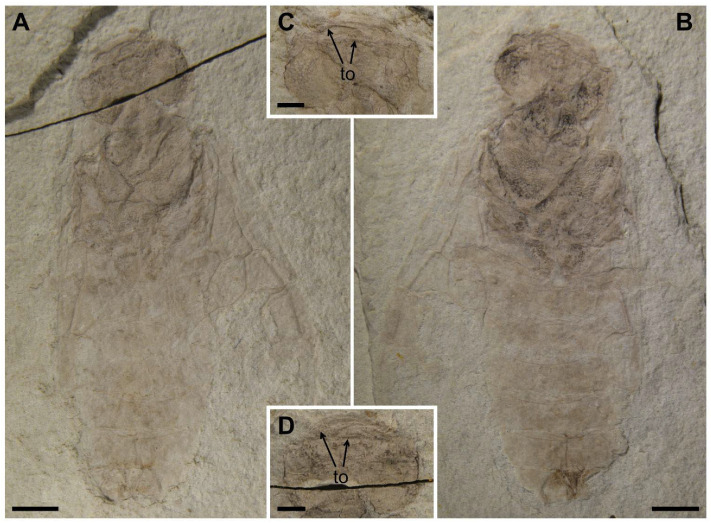
*Scabolyda latusa* sp. nov., Paratype (CNU-HYM-NN2022103p/c) female. (**A**) Dorsal view of body as preserved. (**B**) Ventral view of body as preserved. (**C**) Ventral view of head. (**D**) Dorsal view of head. Symbol: to = tooth. Scale bars: 2 mm in (**A**,**B**); 1 mm in (**C**,**D**).

**Figure 4 insects-13-00402-f004:**
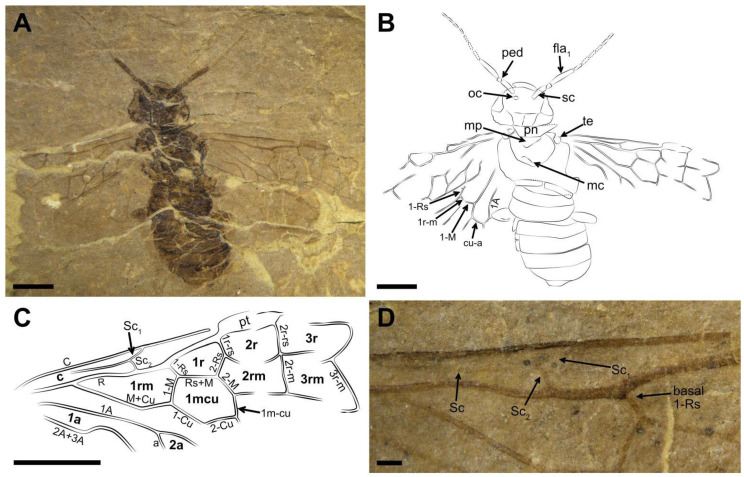
*Scabolyda tenuis* sp. nov., Holotype (CNU-HYM-NN2022104) sex unknown. (**A**) Dorsal view of body as preserved. (**B**) Line drawing of body. (**C**) Line drawing of forewing (partly). (**D**) Photo of Sc in forewing. Symbols: oc = ocellus; pn = pronotum; mp = mesoprescutum; mc = mesoscutellum; te = tegula. Scale bars: 2 mm in (**A**–**C**); 0.25 mm in (**D**).

**Figure 5 insects-13-00402-f005:**
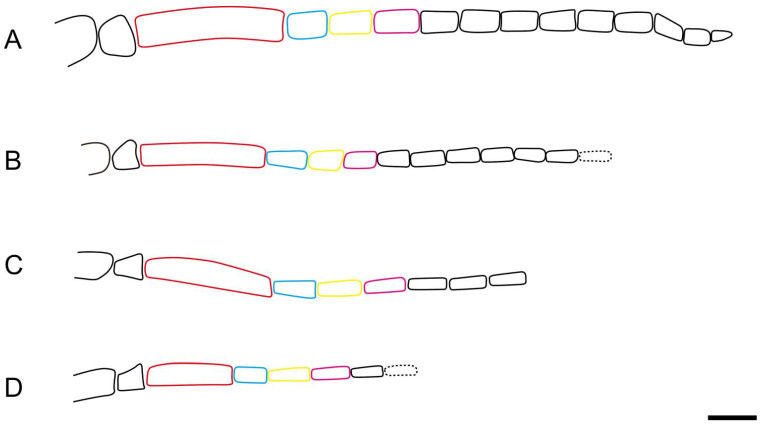
Line drawings of different types of antennae in *Scabolyda*. (**A**) *Scabolyda latusa* sp. nov. (**B**) *Scabolyda orientalis* Wang, Rasnitsyn, Shih and Ren, 2014. (**C**) *Scabolyda tenuis* sp. nov. (**D**) *Scabolyda incompleta* Wang, Rasnitsyn, Shih and Ren, 2014. Notes: flagellomeres in red, blue, yellow, and purple colors = the first, second, third, and fourth flagellomere, respectively. Scale bars: 0.5 mm.

**Figure 6 insects-13-00402-f006:**
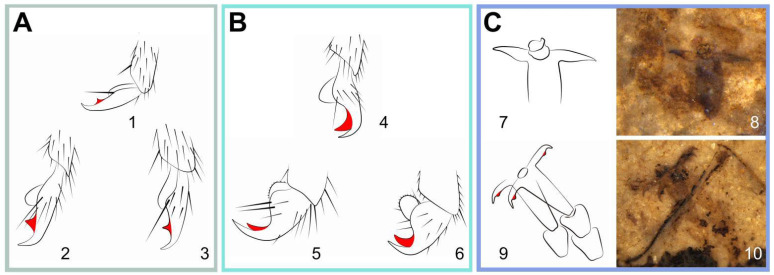
Line drawings of common types of claw in the extant Pamphiliidae modified after Achterberg and Aartsen (1986, Figures 17, 19, 21, 51, 314 and 321) [10] and the extinct *Scabolyda*. Photos 6. C-8 and 6. C-10 were taken wetted with 95% ethanol. (**A**) claws in Cephalciinae: 1, *Cephalcia arvensis* Panzer, 1803; 2, *Acantholyda erythrocephala* Linnaeus, 1758; 3, *Acantholyda teunisseni* Achterberg and Aartsen, 1986. (**B**) claws in Pamphiliinae: 4, *Neurotoma saltuum* Linnaeus, 1758; 5, *Pamphilius alternans* Costa, 1859; 6, *Pamphilius latifrons* Fallen, 1808. (**C**) claws in *Scabolyda*: 7–8, *Scabolyda orientalis* Wang, Rasnitsyn, Shih and Ren, 2014; 9–10, *Scabolyda latusa* sp. nov. Notes: inner tooth in color red.

**Table 1 insects-13-00402-t001:** The ratio of main flagellomeres in *Scabolyda*.

Species	Length Ratio of the First Flagellomere to the Second	Length–Width Ratio of the Second Flagellomere	Length Ratio of the Second Flagellomere to the Third	The Second Flagellomere Whether Obviously Wider than the Third	Length–Width Ratio of the Third Flagellomere
*Scabolyda latusa* sp. nov.	3.9	1.6	0.9	yes	2.3
*Scabolyda orientalis*	3.1	2.5	1.2	no	2.1
*Scabolyda tenuis* sp. nov.	3.0	3.6	1.0	no	3.5
*Scabolyda incompleta*	2.7	2.2	0.8	yes	4.1

## Data Availability

No new data were created or analyzed in this study. Data sharing is not applicable to this article.

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
