# Peer review of "Two New Fossil Sawflies of Pamphiliidae (Hymenoptera: Symphyta) from the Mesozoic of Northeastern China†"

_insects, 2022, doi:10.3390/insects13050402_

Round 1
Reviewer 1 Report
The main question addressed in the research is the description of two new extinct species of Pamphiliidae from the Mesozoic of northeastern China. The biological group in question is extremely important, as it is one of the most basal groups of Hymenoptera, so the discovery of more species of this group, mainly extinct, may clarify the understanding of the evolution and emergence of Hymenoptera. Additionally, the authors provide detailed information about the morphology of the group, which was previously confused and which now allows a connection between the living and extinct lineages.
Few Mesozoic fossils have been reported in Pamphiliidae (only three species until now) and in this MS, two more species are being described, that is, there is an increase of ~65% of species. In addition to the taxonomic descriptions, which alone justify the originality of this MS, the authors explore some minute but important characters used for the classification of extant taxa are difficult to be preserved in reported fossils.
All photographs, drawings and taxonomic aspects are well executed. My only suggestion would be to include a "Conclusions" topic, this would allow the authors to be more specific about the conclusions of this study.
Author Response
Thanks for your kind suggestions and the “Conclusions” part has been added.
Reviewer 2 Report
a nice paper, worth being published, but it is necessary to precise the diagnostic characters and those employed in the key to species
I have a problem, one of your main characters to separate the species is the lenght of 4Rs, supposed to be very short in latusa and quite longer in other species, but, comparing this vein in the different species, this is not obvious at all, what does it mean, it is relative to what ?
and in incompleta 4rs is incompletely preserved ....
the same is for body size, 'relatively small', what does it mean exactly, where do you put large vs. small ?
and the most ratio of flagellomeres is 2-3 vs 3-4, what about when around 3 ?
can you precise all these things ?
and in the diagnoses of the new species, the length of 4rs is not employed, is it normal ?
also in pamphiliids, the first flagellomere is sometime very particular, subsegmented, what about here ?
The character of claw found in the new specimens may suggest the closer relationship 368 between Scabolyda and Cephalciinae, rather than Pamphiliinae
synapomorphy ? can you precise ?
Reviewer 3 Report
I went through the abstract and introduction. There are simply too many grammatical errors (see marked text) to be considered ready for review. Sorry.

Author Response
We revised our manuscript, and the English has been improved. Please read our revised version.
Reviewer 4 Report
This is an interesting work that contributes to the study of Jurassic Pamphiliidae. The requirements of the code of zoological nomenclature are observed. Descriptions are correct. I made some comments in the text. Important. This article is not about the new synonymy and therefore the use of the syn. nov. is strictly prohibited.

Author Response
Referring to the pdf file you gave, we revised the manuscript.
Please read our revised version.
Round 2
Reviewer 3 Report
The manuscript is sound scientifically. The English is still very poor and needs work. I have highlighted text that is faulty, but I do not have the time to correct all of the English language mistakes. So sorry that you have to write in English; hopefully there will come a day when automatic translations are of high quality. The only scientific comment that I have is with the term "spots". I have not heard of this character before. If it is a valid character, I suggest giving a definition and a citation. The manuscript with gramme problems highlighted is attached.

Author Response
Thank you for your valuable comments. We have accepted and adopted nearly all comments and corrections and revised our manuscript accordingly.
The replies to your comments are as follows:
- I am a little confused about your term "spots". Can you be more precise?
Reply: We have replaced it with the word “punctation”, meaning the spots or depressions on the body surface.
- “minor” do you mean smaller?
Reply: Have corrected! Replace it with the word “small”.
- “cell 1mcu”, I am not familiar with this terminology. Should it not be 1m-cu?
Reply: This is indeed talking about the wing cell 1mcu, not the vein 1m-cu. This statement is common in Pamphiliidae.
Correct grammar and the use of technical terms are the foundation of a good article. Thank you again for the careful revisions to the article.